# Determination of the Stability of a High and Steep Highway Slope in a Basalt Area Based on Iron Staining Anomalies

**Lihui Qian** [1,2], **Shuying Zang** [1,3,*], **Haoran Man** [1], **Li Sun** [1,3] and **Xiangwen Wu** [1,3]

1   Heilongjiang Province Key Laboratory of Geographical Environment Monitoring and Spatial Information Service in Cold Regions, Harbin Normal University, Harbin 150025, China

2   Jilin Branch of China National Geological Exploration Center of Building Materials Industry, Changchun 130033, China

3   Heilongjiang Province Collaborative Innovation Center of Cold Region Ecological Safety, Harbin 150025, China

*   Correspondence: zsy6311@hrbnu.edu.cn

**Abstract:** In recent years, geological disasters have frequently occurred on basarlt highway slopes. Studying the stability of highway slopes in this type of area is of great significance for traffic safety. However, due to the high cost and low efficiency of traditional monitoring and experimental methods for slope engineering, these methods are not conducive to the quick and comprehensive identification of regional slope stability. Due to the high iron content of basalt, iron staining anomalies in the ore prospecting field are reinterpreted from an engineering perspective in this study. Taking the S3K section of a highway in Changbai County, China, as an example, Landsat8 remote sensing (RS) images from 2014, 2016, 2018, 2020, and 2021 are selected, and principal component analysis is used to extract iron staining anomalies in the region. Combined with field investigation and evidence collection, the corresponding rock mass fragmentation is distinguished via iron staining anomalies. Then, according to previous research results, eight indexes including annual rainfall, slope, topographic relief, surface roughness, vegetation index, leaf area index (LAI), root depth of vegetation, and human activity intensity are selected for investigation. The artificial neural network–cellular automata (ANN-CA) model is established, and the rock fragmentation classification data obtained based on iron staining anomalies are used to simulate the area. Next, the calculation formula of slope stability is determined based on the simulation results, and the stability of a high and steep slope in the area is calculated and analyzed. Finally, a comparison with an actual field investigation shows that the effect of the proposed method is good. The research findings reveal that it is feasible to judge the stability of a high and steep slope in a basalt area via the use of iron staining anomalies as an indicator. The findings are tantamount to expanding the application scope of RS in practical engineering.

**Keywords:** Landsat8; iron staining anomalies; ANN-CA; slope stability; spatiotemporal evolution

## 1. Introduction

With the development of society in recent years, humans have faced many natural disasters in their living environments, among which geological disasters are common. Geological disasters in basalt platforms have been frequent [1–7] under the combined action of control and inducing factors, so it is particularly necessary to study these characteristic areas. However, the analysis of slope stability has been a challenge for researchers and engineers beginning with Hooke [8] in the 17th century.

Slope stability analysis methods include both qualitative and quantitative approaches. In 2010, Shen [9] summarized slope stability research methods and achievements in detail and studied the stability of highway slopes in southeast Jilin Province, China, through combining mathematical and physical methods. In recent years, research on the stability of basalt slopes has mainly focused on the use of quantitative analysis methods. For example, Zhou [10] and Li [11] studied the slope stability in a Guizhou basalt area via a field survey

and the finite element method and analyzed the process of the influence of water infiltration on slope stability. In 2015, Kainthola [12] studied the stability of the Mahabaleshwar basalt slope in India through combining experimentation and a numerical model. In 2018, He [13] used Midas GTS numerical analysis software to study the slope stability of the new village of the Baihetan Hydropower Station. In 2021, Liu et al. [14] proposed a simplified dual-strength finite element method and used the ABAQUS software platform for secondary development to obtain a method consistent with the stability analysis of weathered basalt soil mass in heavy rainfall areas. In the same year, Zhang [15] obtained the change rule of rock mass strength under different water contents via triaxial test analysis and investigated the safety and stability of basalt slopes with ABAQUS finite element software. In addition, some new technologies have been continuously applied in practical applications and good results have been achieved. For example, helicopter remote sensing (RS) technology has been used to identify rock mass strength [16], and high-resolution three-dimensional point clouds have been collected to analyze slope stability [17,18].

The previously mentioned schemes, geotechnical investigation, and multiple models can be selected for the calculation and analysis of slope stability in basalt areas. [19,20]. However, although these methods are feasible, if regional slope stability research is conducted, the workload will be increased dramatically, and personnel with high theoretical and technical skills will be required. The implementation would be not only difficult and time-consuming, but also very uneconomical. In fact, studies show that the mechanical properties of a rock mass are closely related to its structural plane, structural body, and occurrence environment [21]. Therefore, various geological interfaces in a rock mass, such as folds, faults, beddings, joints, and schistosity, can lead to a change in the mechanical properties of the complex natural environment; thus, the rock mass can be gradually broken, which renders it vulnerable to water infiltration. When water acts on the rock mass, the small material particles existing in the fracture can migrate through the hydraulic force via the actions of corrosion, abrasion, and erosion, which will increase the porosity of the rock mass [22]. With the passage of time, this is likely to cause instability in rock masses of high and steep slopes, and even disaster.

Rock mass alteration generally occurs under a peculiar geological structural background [23]. Some studies have demonstrated that rock mass alteration will lead to a decrease in mechanical properties [24,25]. The differences caused by alteration can easily cause the heterogeneity of the rock mass properties and can form a weak zone of the engineering rock mass in the local area [26]. Rock mass alteration has a definite causal relationship with the weathering, crushing, and mechanical properties of the rock [27]. In other words, rock mass alteration will cause slope instability. Iron staining anomalies are a type of alteration that reacts to environmental changes and is more noticeable in basalt areas with high iron content. In other words, when the detritus on the surface of a basalt slope rock mass with high iron content migrates spatially in the natural environment, the iron film cannot develop in time, which will lead to different spectral information at the same spatial location. This relationship can be used for identification based on RS images.

As Qian Xuesen stated, "the best way to study things on the ground is in the sky. It is very time-saving and the results are accurate" [28]. In view of this, the use of RS technology for iron staining anomaly recognition is mature [29]. The S3K section of the Yalu River Highway in Changbai Korean Autonomous County of Jilin Province, China, was chosen as the research object of the present study. The area is located on a volcanic platform, which is mainly basalt with high iron content [30,31], and the rock fracture development of the highway slope is characterized by strong water action [32]. This has resulted in a corresponding fracture surface with a high degree of thin iron film development, and it is easy to identify abnormal changes in iron staining from RS images. Moreover, the geological disaster points of the S3K highway section are concentrated, and the highway is the artery of the local economy; thus, it has an important strategic position and affects national geopolitical security [33,34]. Finally, this area is a typical volcanic platform, and has a certain representative status.

During the research process, principal component analysis (PCA) is used to extract iron staining anomalies, and the degree of rock fragmentation is categorized according to the field survey data. Referring to previous research results [29,33–37] and on the basis of water–rock reaction theory, the index factors affecting slope instability (see Section 4.4) are selected. Considering that the cellular automata (CA) model has the ability to simulate the spatiotemporal evolution process of complex systems, it is easier to implement than other models [38]. The combination of the artificial neural network (ANN) and CA models can achieve the dual advantages of accuracy and speed. Thus, the ANN-CA model is established to simulate the study area. Finally, Equation (11) is proposed to evaluate the stability of a regional basalt slope with iron staining anomalies as an indicator (see Section 5). After field verification, this method is determined to be suitable for this characteristic area.

## 2. Study Area

Changbai County is located in the southeast region of Jilin Province, China, at the southern foot of Changbai Mountain and the right bank of the upper reaches of the Yalu River. Its geographical coordinates are 127°12′20′′–128°18′10′′ east longitude and 41°21′41′′–41°58′02′′ north latitude. It is the most important Quaternary volcanic rock distribution area in China, with a total area of 2497.6 km$^2$. The territory is characterized by dense mountains with peaks and valleys. Moreover, the traffic is relatively closed and is mainly highway traffic.

The topography of the area is complex, with high terrain in the northeast and low terrain in the southwest. The highest altitude is about 2457.4 m, the lowest altitude is about 450 m, and the average altitude is about 1570 m. The S3K section is located on the northern side of the Yalu River, which is a slope zone in the front of the basalt platform. The slope is steep with a relative height difference of about 300 m. There are 10 mountains in the area where high-risk collapse geological disasters occur frequently. The general situation of the area is illustrated in Figure 1.

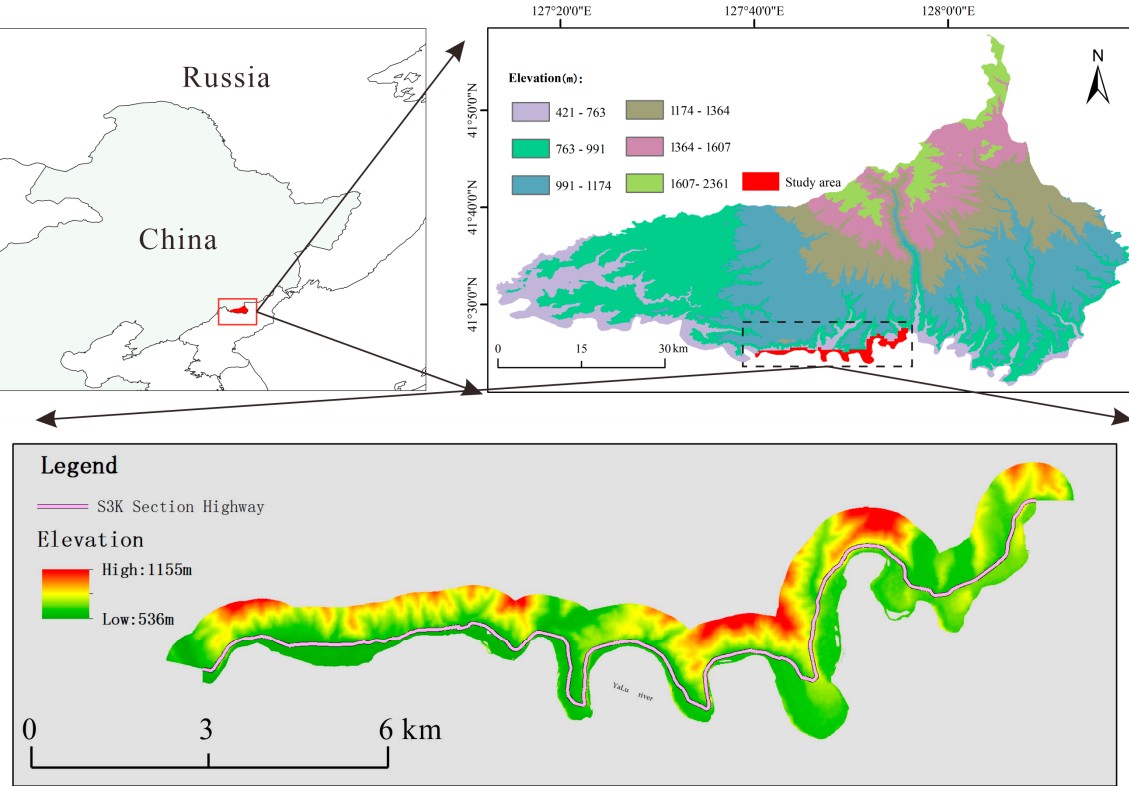

**Figure 1.** The map overview and location of the S3K section (Qian et al. [36]).

## 3. Theory and Methods

### 3.1. CA and ANN

The CA model is a grid dynamics model with discrete time, space, and state characteristics, as well as local spatial interaction and temporal causality. It has the ability to simulate the spatiotemporal evolution process of complex systems [39]. It is an idealized model of a physical system whose physical parameters only include a finite number of sets. The standard CA model is expressed in mathematical terms as follows:

$$A = (Z_n, S, N, f) \tag{1}$$

where $A$ represents a CA system, $Z_n$ represents $n$-dimensional Euclidean space, $n$ is the dimension of the cellular space, and $S$ is the finite discrete state set. $S = \{S_1, S_2, S_3, \ldots S_i \ldots S_k,\}$, among which $S_i$ denotes state $I$ of the CA. Moreover, $N$ is the neighborhood of the central cell, which is a finite sequence subset of $S_i$, $N = (x_1, x_2, \ldots, x_i, \ldots x_n)$. Finally, $x_i$ is the position of adjacent cells relative to the central cell, and $f$ is the evolution rule of $S$ from time $t$ to time $t + 1$. The structure of the CA is shown in Figure 2.

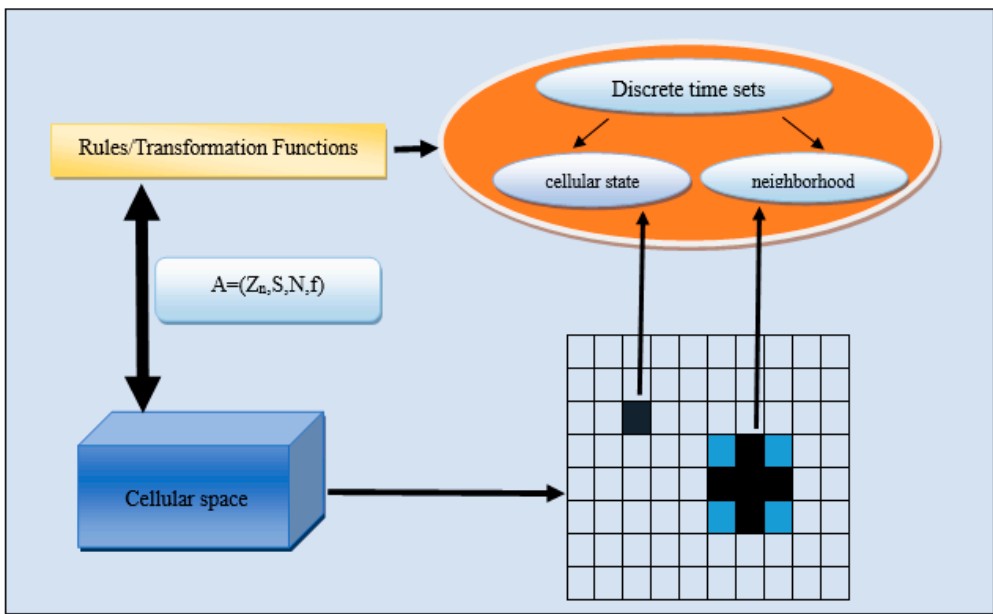

**Figure 2.** The structure of the CA model (Zhou [40]).

The ANN is a nonlinear mathematical model that imitates the structure and function of biological neural networks. It is composed of a large number of nodes and connections between nodes. These nodes are called neurons, which are also cells in the human brain. The construction of neural networks is the creation of machines that can simulate the brain to realize artificial intelligence. As shown in Figure 3, each neuron contains one or more dendrites. Dendrites are the input nerves of neurons that receive information from other neurons. Each neuron also contains an axon, which is the output nerve of a neuron that is used to transmit information to other neurons. Neurons have two working states, namely, excitation and inhibition. Usually, neurons are in the state of inhibition. When the input signal of the dendrite reaches a certain level, neurons change from inhibition to excitation, and axons send signals to other neurons. Based on this principle, a neuron can be regarded as a computing unit of the brain, and a neural network composed of neurons can be regarded as a model simulating the brain. Neurons are divided into an input layer, a hidden layer, and an output layer. The neural network receives the original feature information through the input layer, then processes and extracts the feature information through the hidden layer, and outputs the results through the output layer. The application model was constructed in combination with the CA model, as detailed in Section 3.3.

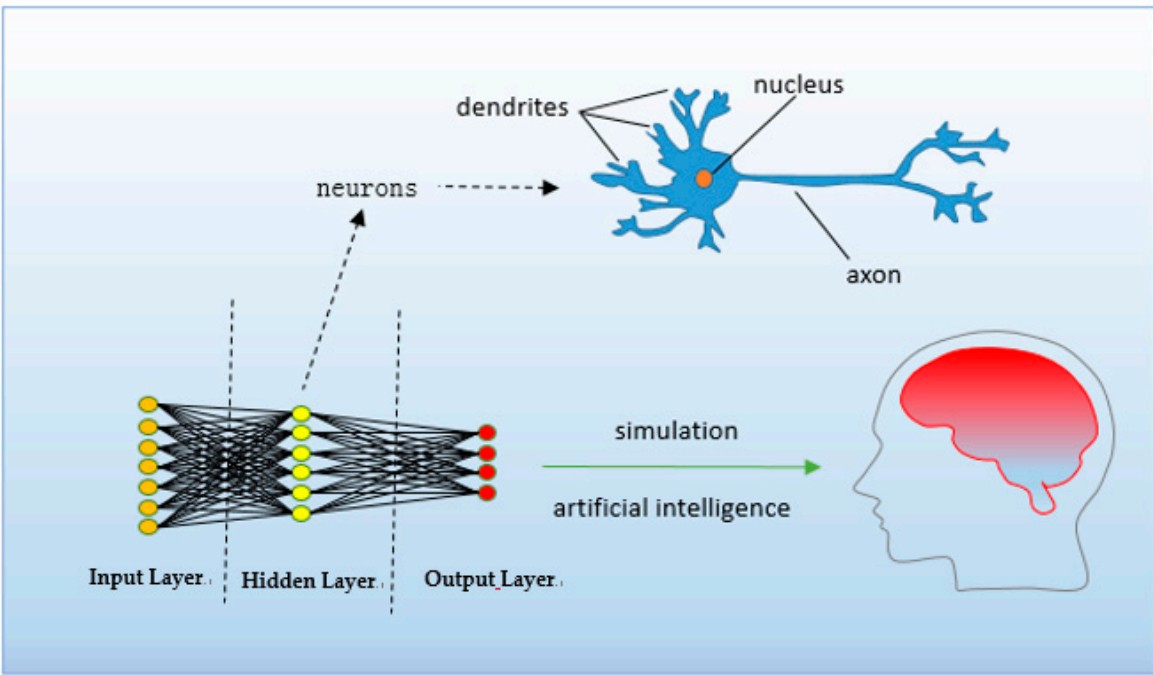

**Figure 3.** The principle and structure diagram of the ANN.

### 3.2. Theoretical Basis

The formation and evolution of the slope surface morphology have both spatial and temporal characteristics [41]. Rock mass alteration influences the change of the rock mass strength, which leads to rock mass fragmentation. According to the erosion cycle theory put forward by Davis [42], the broken surface debris of a high and steep slope will migrate in space under the action of gravity and many other factors, and will then re-expose the original rock surface.

According to the pertinent results of RS research [29,43], a rock mass exists in one of four states, namely, complete, relatively broken, broken, and extremely broken, which are represented by 0, 1, 2, and 3, respectively. Under the action of gravity and other factors, the surface debris of a sloping rock mass changes in the following ways: $3 \rightarrow 2$, $3 \rightarrow 1$, $3 \rightarrow 0$, $3 \rightarrow 1$, $2 \rightarrow 0$, and $1 \rightarrow 0$. The overall evolutionary process of the slope occurs based on the cycle of instability–stability–instability–stability, which repeatedly affects the change of the alteration information in both time and space.

In this study, each cell represents the fragmented situation of the slope where it is located. According to previous research [30–33,37,43–46], RS technology [29,47] is used to extract the iron staining anomaly value of the slope, and, in combination with field verification, the cell structure is simplified and expressed by 0, 1, 2, or 3, respectively (see Section 4). The cell radius of different rock masses may be different, and it is difficult to determine. The Moore model has eight neighbors, i.e., the life and death of a cell are determined by its own state at a certain moment and the state of its eight neighbors, which is consistent with the law of rock mass evolution. Thus, the Moore type is selected as the neighbor type of the CA, and the neighborhood is invoked as the scope of the evolution rule. Furthermore, the advantages of the ANN, namely, that it can effectively deal with noise and redundant or incomplete data, are considered [38]. The ANN is especially suitable for dealing with nonlinear or complex geological systems that cannot be characterized using mathematics. Thus, it is combined with the CA model to establish the ANN-CA model. The research process is presented in Figure 4.

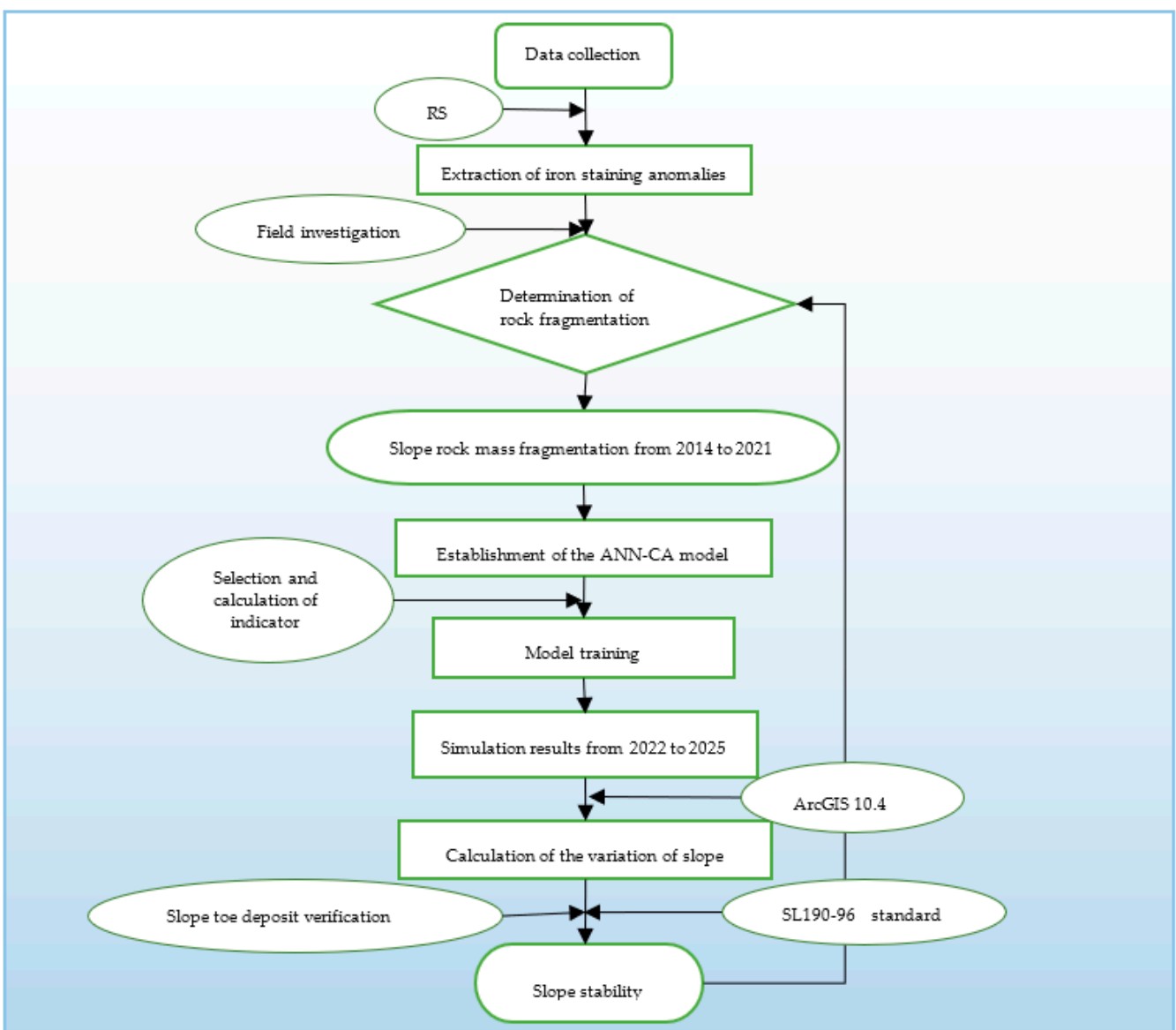

**Figure 4.** The technique flow chart.

*3.3. Model Implementation Process*

First, according to relevant studies [48–51], the neurons in the hidden layer are called current neurons, which receive the signals of the neurons in the input layer. The selected indexes affecting the slope stability (see Table 3) were normalized via ArcGIS10.4 software. The sequence is $X = [X_1, X_2, X_3, \ldots, X_i]$, and it was assumed that their corresponding weights are $W = [w_1, w_2, w_3, \ldots, w_i]$. They were connected with the current neurons to achieve the purpose of transmitting information.

Then, the expression of the set formed by the cell $K$ at the simulation time $t$ can be expressed as Equation (2), where $T$ represents the transpose operation.

$$X(K, t) = [x_1(k,t), x_2(k,t), x_3(k,t), \ldots x_i(k,t)]^T \tag{2}$$

Second, the hidden layer multiplies the received $X(k, t)$ sequence by the corresponding weight $W$ and sums the results. Thus, the signal received by the current $J$-th neuron can be recorded as $net_j(k, t)$ (Equation (3)).

$$net_j(k, t) = \sum_j w_{j,i} X(k, t) \tag{3}$$

Third, the input signal is activated by the hidden layer, and the output of the neuron is finally obtained. The conversion probability of the output layer value is shown in Equation (4).

$$P(k, t, l) = (1 + (-\ln \gamma)^{\alpha}) \times \sum_j w_{j,l} \frac{1}{1 + e^{-net_j(k,t)}} \tag{4}$$

In this equation, $1 + (-\ln \gamma)^{\alpha}$ represents the random disturbance factor, $\gamma$ is a random value within the range of [0, 1], and $\alpha$ controls random disturbance and expresses the uncertainty caused by random factors in the process of rock mass evolution; in this study, its value was set within the interval [1, 10]. Finally, $\frac{1}{1+e^{-net_j(k,t)}}$ is the response value of the hidden layer.

Because error correction is not carried out in these three steps, it is difficult to meet expectations. Thus, it is necessary to train the model. In this process, the neural network compares the output with the actual output to clear the error, which implies the direction and amplitude of the prediction results to be adjusted later. This information is backpropagated by the neural network, and the weight is adjusted for multiple training, which continues until the weight value can fit the simulation prediction (see Figure 5).

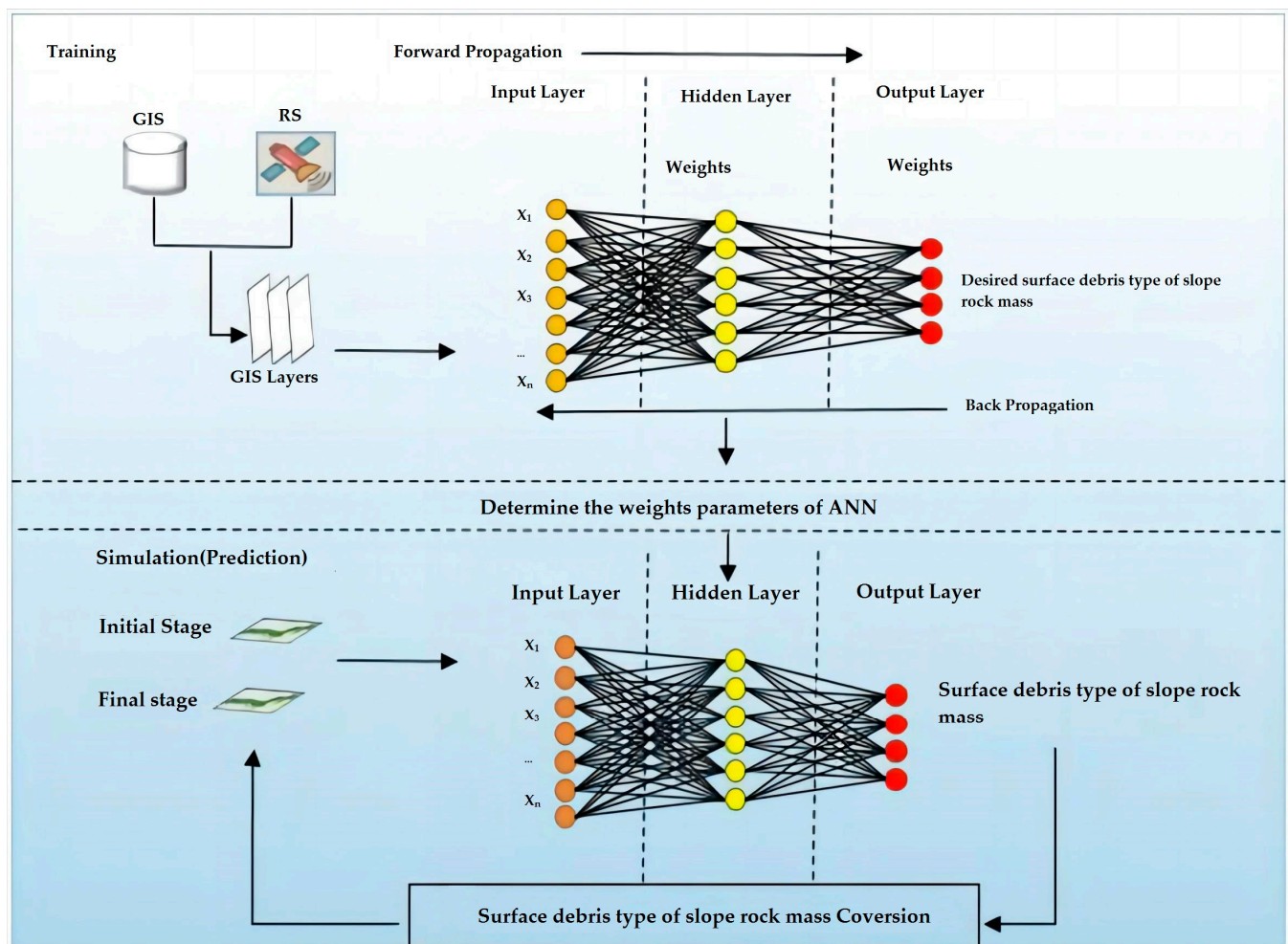

**Figure 5.** The schematic diagram of the ANN-CA establishment and implementation process.

To better understand the implementation process of ANN-CA, the computer program must be supplemented. First, the data layer of each factor affecting the change of the rock mass structure type (see Table 2) and the current rock mass structure grid layer were used as input data (see Section 4.2 for the explanation of the acquisition of this data). Then, the sampling ratio of the data and the Moore neighborhood were set. The sampling ratio

must be repeatedly adjusted according to the simulation results. Next, after completing the first two steps, the ANN parameter settings were entered, including the learning rate, the proportions of training and verification data, the number of cells in the hidden layer, the number of iterations, etc. Then, the settings of simulation data were entered, and the change of the rock mass structure from the beginning to the end was considered the total simulation amount. When the total amount was reached, the program was terminated. In the program, the quotient of the total conversion amount and the number of iterations was taken as the number of cells for each iteration, and the cells for each iteration were randomly selected by the program. If it could be converted, the number of conversions was increased by 1. When the number of iterations was reached, the next iteration was executed. Finally, the accuracy was checked, and if the accuracy was not satisfied, the training parameters were adjusted until the requirements were met. Next, the conversion matrix (see Section 3.2 for the rules), disturbance coefficient, conversion threshold, etc. (see Section 4.4.2), were determined. Finally, the relevant model parameters were repeatedly adjusted according to the accuracy of the simulation results until the research was satisfied.

## 4. Data Processing and Simulation

### 4.1. Image Acquisition and Preprocessing

Because the Operational Land Imager (OLI) sensor has a narrow band range as compared with the Thematic Mapping (TM+) sensor, it can better extract alteration information [47]. In combination with historical meteorological data, Landsat8 RS images with a cloud volume of less than 2% and dated 30 May 2014, 19 May 2016, 18 October 2018, 29 May 2019, 30 May 2020, and 18 June 2021, respectively, were downloaded from the Geospatial Data Cloud (https://www.gscloud.cn, accessed on 5 June 2023). The 1:10,000 Digital Elevation Model (DEM) data were sourced from the surveying and mapping department, domestic rainfall data were collected from the meteorological department, and land-use data were obtained from the local natural resources bureau.

### 4.2. Extraction of Iron Staining Abnormalities

First, geometric correction, radiometric calibration, and FLAASH atmospheric correction of the RS images were performed with ENVI5.2 software. The RS images were then masked with the vector data of the S3K segment after coordinate registration. According to the existing literature [29], the main information of iron oxides is concentrated in bands 2, 4, 5, and 6, and the seventh band is omitted to avoid the influence of hydroxyl and carbonate minerals. Thus, the combination of bands 2, 4, 5, and 6 was selected for the PCA of iron staining anomalies. For the convenience of illustration, only the PCA tables for 19 May 2016 and 30 May 2020, are reported, as shown in Table 1. The other time-image data processing methods were similar.

**Table 1.** The statistical analysis of the principal components of iron staining anomalies.

| Date | PC | Band 2 | Band 4 | Band 5 | Band 6 | Contribution Rate (%) |
|------|----|--------|--------|--------|--------|----------------------|
| 19 May 2016 | PC1 | 0.094610 | −0.098194 | 0.989432 | 0.049301 | 76.49 |
| | PC2 | −0.617834 | −0.773717 | −0.010745 | −0.139745 | 22.33 |
| | PC3 | −0.646450 | 0.398389 | 0.069112 | 0.647002 | 0.94 |
| | PC4 | 0.437531 | −0.482707 | −0.127011 | 0.747950 | 0.24 |
| 30 May 2020 | PC1 | 0.195103 | −0.033706 | 0.978984 | 0.048875 | 77.42 |
| | PC2 | 0.674310 | 0.720196 | −0.115350 | 0.115408 | 21.45 |
| | PC3 | 0.646498 | −0.540702 | −0.121278 | −0.524379 | 0.86 |
| | PC4 | 0.298801 | −0.433386 | −0.116516 | 0.842211 | 0.27 |

Based on the PCA results (see Table 1), and considering the spectral characteristics of iron-stain-altered minerals [29], component PC1 was selected for analysis. According to the histogram curve statistics provided via ENVI software, the digital number (DN) value of the principal component layer in each year was calculated.

### 4.3. Fragmentation Classification of the Slope

The distribution layer of iron staining anomalies obtained via PCA was correlated with the spatial position of the survey data, and the DN value range corresponding to the slope breakage was obtained, as shown in Table 2.

**Table 2.** The comparison between the measured data and abnormal DN values of iron staining anomalies.

| No. | Longitude and Latitude of the Center of the Survey Area | Investigation of the Slope Rock Mass | DN Value Range |
|---|---|---|---|
| 1 | 127°55′45″, 41°27′27″ | Broken to extremely broken | 224~255 |
| 2 | 127°50′46″, 41°25′11″ | Strongly weathered to weakly weathered, and relatively broken | 202~255 |
| 3 | 127°48′05″, 41°25′18″ | Strongly weathered to weakly weathered, and relatively broken | 190~255 |
| 4 | 127°46′53″, 41°25′23″ | Completely weathered to strongly weathered, and broken to extremely broken | 196~255 |
| 5 | 127°40′33″, 41°25′12″ | Strongly weathered to weakly weathered, and relatively broken | 200~255 |
| 6 | ---- | Weakly weathered, and relatively complete | 88~255 |
| 7 | ---- | Relatively broken | 187~255 |
| 8 | ---- | Relatively complete | 90~255 |
| 9 | ---- | Extremely broken | 242~255 |
| 10 | ---- | Broken | 197~255 |

Table notes: ① Please refer to the extant literature [31] for details on the slope data sources for Nos. 1–5. ② The slope data for Nos. 6–10 were obtained from field surveys. ③ The longitude and latitude coordinate data are classified and not listed in the table.

According to Table 2, the other 70 slope rock mass structures in the region were investigated and verified (as shown in Table A1), and repeated numerical experiments were carried out. Finally, DN values of less than 40 were set as "other types", values of 40–130 were set as "complete", values of 130–220 were set as "broken", values of 220–245 were set as "relatively broken", and values greater than 245 were set as "extremely broken". Corresponding to Table 2 and Schedule 1 are integral block, block structure, fragmentation structure, and granular structure. The reclassification tools in ArcGIS10.4 software were respectively set to 0, 1, 2, and 3. The classification results from 2014 to 2021 are plotted in Figure 6.

From Figure 6, it is evident that there were noticeable changes in the distribution of iron staining anomalies in the slope from 2014 to 2021, thus reflecting spatiotemporal variation. The most direct reason for the change was extremely heavy rainfall. Therefore, the iron staining anomaly map obtained from RS analysis on 18 June 2021, after extreme weather, was greatly changed as compared to that from 18 October 2018. For example, in May and July 2018, continuous heavy rainfall in the region led to the instability and collapse of multiple slopes, and rainy weather in July 2020, as well as extreme rainstorm weather (https://m.thepaper.cn/newsDetail_forward_8985114, accessed on 8 June 2023) caused by typhoons Bawei and Meissack in September of that year, led to the large-scale erosion of highway slopes. Further research was therefore carried out, as subsequently presented.

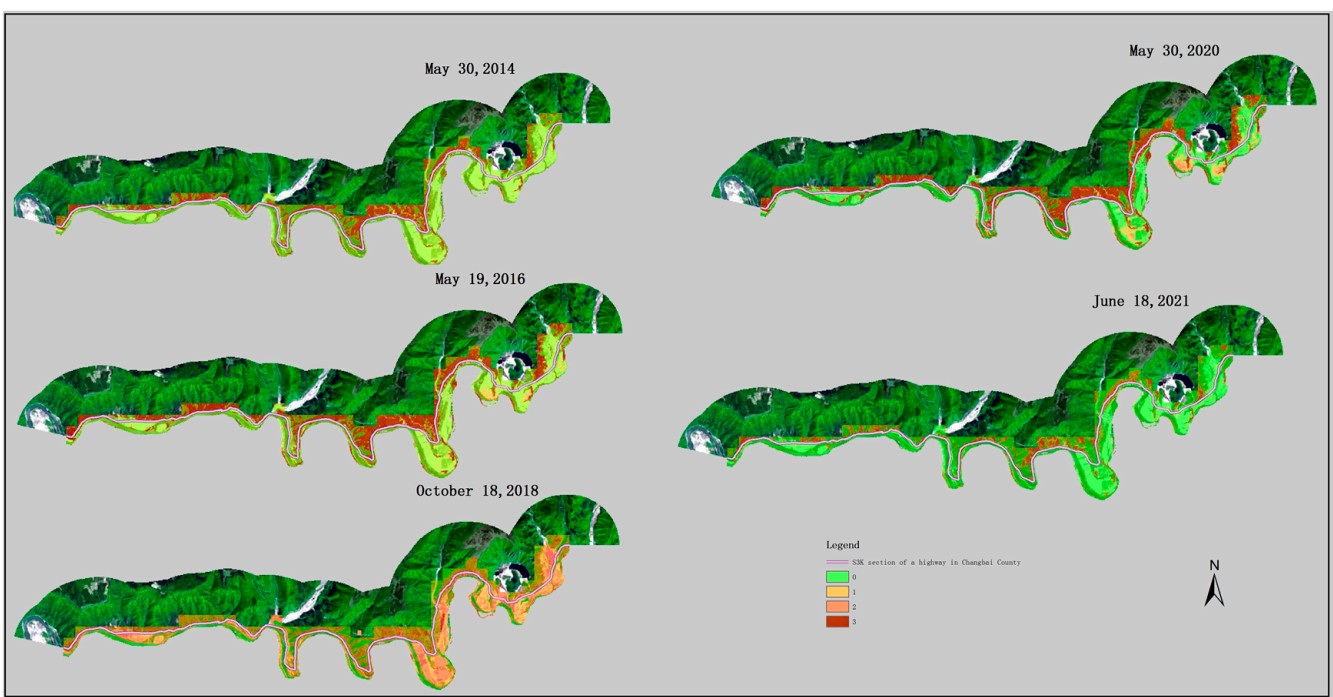

**Figure 6.** The schematic diagram of the degree of rock fragmentation from 2014 to 2021 based on iron staining anomalies.

### 4.4. Simulation of Slope Instability Evolution

#### 4.4.1. Indicator Selection and Acquisition

The First Law of Geography holds that geographical objects or attributes are interrelated in terms of spatial distribution, which includes clustering, random, and regular distributions. As mentioned in Sections 1 and 3.2, the instability of the slope is closely related to rock fragmentation, and this can be reflected in the change of iron staining anomalies.

Referring to the research results of Bell [52] on the relationship between geological disasters and rainfall, as well as the literature on geological disasters in the region [27,28,30,41], it is agreed that topography and geomorphology have important impacts on slope stability; thus, slope, topographic relief, and surface roughness were selected for analysis. Second, the leaf area index (LAI) can reflect the balance of the surface energy and interact with many factors of precipitation [53]. Rock and soil systems are complex open systems characterized by the exchange of energy and matter [54,55]. Thus, the instability of the slope is manifested as an exchange of energy. Third, the root depth of slope vegetation affects not only the absorption of soil moisture, but also the soil holding capacity, and a larger root depth decreases soil erosion and instability. Finally, human engineering activities lead to a decrease in the mechanical properties of slope rock mass, resulting in instability. The control factors of the ANN-CA model were determined as the rainfall, slope, topographic relief, surface roughness, vegetation index, LAI, root depth of vegetation, and human activity intensity.

First, the Inverse Distance Weighted (IDW) tool module in ArcGIS10.4 software was used to interpolate the rainfall space to obtain the rainfall layer data.

Then, the slope was obtained using the slope tool in ArcGIS10.4 software, and the topographic relief was extracted from the slope data layer based on the 1:10,000 topographic data of the region. Moreover, the surface roughness was calculated according to Equation (5). The topographic relief was calculated using the Range module in ArcGIS10.4

software to calculate the maximum and minimum values of pixels in the neighborhood, and the difference between them was obtained.

$$R = \frac{1}{\cos(slope \times \frac{3.14159}{180})} \tag{5}$$

Third, while there are many calculation methods for the *LAI*, the cubic polynomial regression equation was used in this study, as given by Equation (6):

$$LAI = 14.544 \times NDVI^3 + 1.935 \times NDVI^2 - 3.877 \times NDVI + 1.798 \tag{6}$$

where *NDVI* is the normalized difference vegetation index, the calculation of which is obtained using

$$NDVI = \frac{r_{NIR} - r_R}{r_{NIR} + r_R} \tag{7}$$

where $r_{NIR}$ is the near-infrared band and $r_R$ is the infrared band.

Fourth, based on the collection of field vegetation data for section S3K and the comparison of *LAI* data obtained in the third step, it was found that the *LAI* value corresponding to vegetation with a root depth over 60 cm exceeded seven. For the sake of research precision, Yang [56] suggested that the maximum *LAI* values of woodland, grassland, and sparse vegetation are 3, 2.6, and 1.0, respectively, which is due to the different geographical and spatial environments and vegetation types in the studied area. The root depth data layer can be calculated according to Equation (8).

$$Rd_i = Rd_{\max}\frac{LAI_i}{LAI_{\max}} \tag{8}$$

Finally, the human activity intensity index was calculated using Equation (9) [57]:

$$DT = \sum_{i=1}^{N} \frac{A_i P_i}{TA} \tag{9}$$

where *DT* is the intensity of human activities, *N* is the number of landscape types, $A_i$ is the total area of landscape component *i*, $P_i$ is the artificial influence intensity parameter reflected by landscape component *i*, and *TA* is the total area of the landscape.

The control factor layer was normalized using the Fuzzy Membership module in ArcGIS10.4 software, and the normalized raster layer was obtained. The datum was used as the influencing factor of the CA model (see Table 3).

**Table 3.** The slope stability control factor used in the ANN-CA model.

| No. | Control Factor | Acquisition Method | Original Data Value Range | Standardization Scope |
|---|---|---|---|---|
| 1 | Annual rainfall | IDW | 622–699 mm | 0~1 |
| 2 | Monthly extreme rainfall | IDW | >200 mm | 0–1 |
| 3 | Slope | Slope tool in ArcGIS10.4 software | 0~81.28° | 0~1 |
| 4 | Topographic relief | Max-min | 0~80 | 0~1 |
| 5 | Surface roughness | Equation (5) | 1–6.15 | 0–1 |
| 6 | LAI | Equation (6) | −6.9–14.4 | 0–1 |
| 7 | NDVI | Equation (7) | −1–1 | 0–1 |
| 8 | Rd | Equation (8) | 0–70 | 0–1 |
| 9 | DT | Equation (9) | 0.83~0.9 | 0~1 |

4.4.2. Model Training and Simulation

First, if the conversion probability $p$ of slope fracture type $L$ at time $t$ is a random factor, then the product of the ANN calculation probability, the neighborhood development density, and the conversion suitability can be expressed as follows [51]:

$$P(k,t,l) = (1 + (-\ln\gamma)^\alpha) \times p_{\text{ANN}}(k,t,l) \times \Omega_k^t \times con(s_k^t)) \tag{10}$$

where $1 + (-\ln\gamma)^\alpha$ is a random factor, $p_{\text{ANN}}(k,t,l)$ is the conversion probability of the slope fragmentation type calculated using the trained ANN, and $\Omega_k^t$ is the neighborhood development density of the defined neighborhood window. Finally, $con(s_k^t)$ is the conversion suitability between the two types, the values of which are 0 and 1, which respectively represent convertible and non-convertible.

In the simulation of normal climate years, the control index factor (Table 3) was set, and the classification results of rock mass fragmentation on 30 May 2014, and 19 May 2016 (see Section 4.2) were used as the data to extract the conversion rules. The sampling ratio was 10% and the neighborhood was 15 m × 15 m. In the model, the integrity of the rock mass was used as a condition for terminating the computer program cycle.

According to geological law, we assume that the rock mass rules were set based on the non-jump principle, e.g., a rock mass with integrity can be converted to a block rock mass, and a non-jump rock mass can be converted to a fragmentary or granular rock mass. In this process, the conversion can be set to 1 and the non-conversion can be set to 0. After setting the rules, 19 May 2016 was set as the starting time, 2020 was set as the end year, and the control index is provided in Table 3. The parameters were repeatedly run and adjusted throughout the simulation experiment. Finally, the disturbance coefficient was 2, the number of iterations was 200, the number of hidden layer cells was 18, the machine learning rate was 0.06, and the conversion threshold was 0.6. The accuracy of the training data set was 90.294%, and the accuracy of the verification data set was 89.324%. The simulation data of a normal year were obtained using this set of parameters to simulate the future (Figure 7a).

Similarly, for extreme years, the classification results of rock fragmentation on 19 May 2016 and 18 October 2018 were used as the extraction and conversion rule data. The model was repeatedly run and the parameters were adjusted. Finally, the disturbance coefficient was 8, the number of iterations was 300, the number of hidden layer cells was 22, the machine learning rate was 0.05, and the conversion threshold was 0.8. The accuracy of the training data set was 82.232%, and the accuracy of the verification data set was 79.648%. The simulation data of abnormal weather years were obtained using this set of parameters to simulate the future (Figure 7b).

Because future environmental change is unknown, the research process was divided into two situations for simulation, namely, normal and extreme weather. The abnormal iron staining value of the slope was found to remain stable under normal conditions, and iron staining anomalies were found to undergo very strong changes in extreme ecological conditions. When the environment tended to be stable, the iron staining anomalies in the same spatial location were found to recover in about two years.

According to the real iron anomaly value in 2021 (an environmentally abnormal year), assuming that 2022 is a normal year, via repeated computer experiments, the disturbance coefficient of the model was determined to be 2, and the conversion threshold was set to 0.6. Assuming that 2022 is an extreme weather year, the disturbance value of the model was 8 and the conversion threshold was set to 0.8. The iron staining anomalies under the two environmental conditions in 2022 were simulated. Similarly, based on the data obtained from considering 2022 as an extreme year, it was assumed that 2023 will be a normal year, and a simulation was carried out. To save space and facilitate discussion, only two simulation results for 2022 are reported (Figure 7). The simulations for 2023, 2024 and 2025 were similar to those for 2022, and are therefore not described.

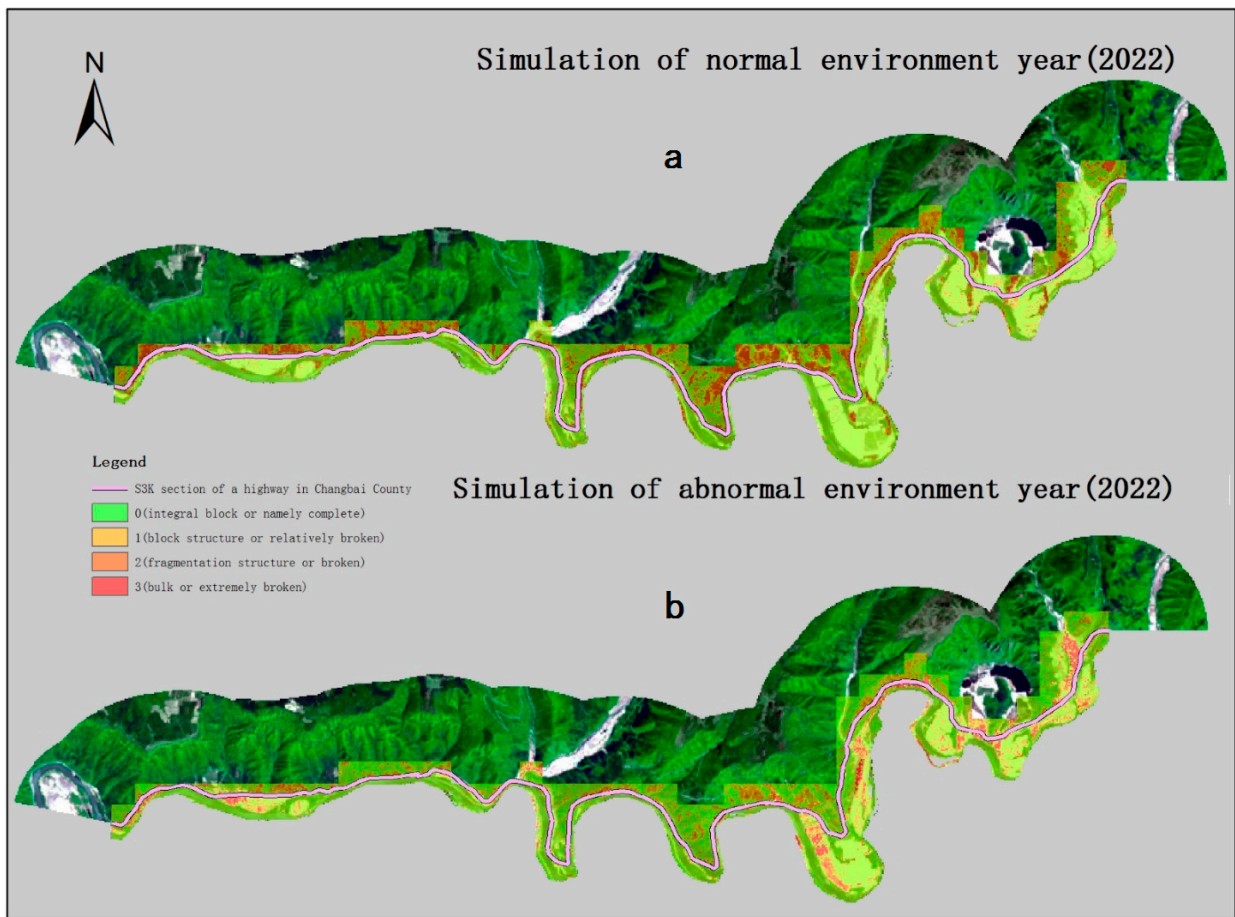

**Figure 7.** A sketch of rock mass fracture under two natural environmental conditions simulated with iron staining anomalies as an indicator. (**a**,**b**) respectively for the explanation of the paper.

## 5. Analysis and Discussion

Deming proposed that the greater the slope, the more serious the erosion, and the corresponding RS spectral characteristics of ground objects will also change significantly [58]. Yanjun posited that the rock mass structure is not static, and that it will change with environmental impacts [59]. It is precisely because of this change that high and steep rock mass slopes are unstable, so further analysis and discussion are needed to determine the stability and evolution law of this type of slope.

### 5.1. Analysis Process

Referring to previous research results [60–64], the concept of an RS-based dynamic index of the instability of highway slopes with iron staining anomalies as the indicator is proposed (e.g., Equation (11)). In other words, the change area of the iron staining anomalies of the slope in the grid serves as the basis for the evaluation of slope instability.

$$I_{TRYC} = \sum_{j-i} \left| \frac{K_j - K_i}{A} \right| \times \frac{1}{T} \times 100\%, (j \leq 3, i \leq 3, j > i, j, i \in N) \qquad (11)$$

where $I_{TRYC}$ denotes the intensity index of unstable variation in the unit, and $K_j$ and $K_i$ respectively denote the areas of iron staining anomalies in the spatial unit at the end and early stages. Moreover, $A$ denotes the grid area, which is a 500 m × 500 m grid established in ArcGIS10.4 software and clipped by the vector layer of the S3K section area, and $T$ represents the time interval between the end and the beginning of the study. The absolute value symbol indicates that whether the calculated value is positive or negative is not considered, and the value is unified as the change intensity.

According to Equation (11), to save space, only the operation process of 2014 and 2016 is described, and the simulation data of 2016 and 2018, 2018 and 2020, 2020 and 2021, and 2021 and subsequent years are successively analyzed.

First, two RS images of iron staining anomalies were classified according to the gray value, namely 0, 1, 2, and 3, which respectively correspond to complete, relatively broken, broken, and extremely broken, for different calculations. Then, they were combined with the Combine tool in ArcGIS10.4 software, and the corresponding image spots of $3 \rightarrow 2$, $3 \rightarrow 1$, $3 \rightarrow 0$, $2 \rightarrow 1$, $2 \rightarrow 0$, and $1 \rightarrow 0$ were extracted. The area was counted and saved as the corresponding layer file using the Geometry Attributes module in the software, and these layers were fused into a layer with the Merge tool. Finally, the grid data were loaded, and the Join tool was selected. The area of the grid occupied by the patches was counted according to the spatial location.

According to the Chebyshev inequality theory and with reference to the standard soil erosion intensity classification table (SL190-96), the statistical data of the slope stability classification in this area were obtained according to the grid ratio of abnormal iron staining changes in normal and abnormal years, as shown in Table 4.

**Table 4.** The statistics of the abnormal change area of iron staining anomalies and the dynamic degree of slope stability.

| Time | Abnormal Change Area of Iron Staining Anomalies (km²) | Ratio of Iron Staining Abnormalities Indicating Change to the Grid Area | Corresponding Historical Disaster Points (Number of Places) | Stability Assessment | Spatial Location |
|---|---|---|---|---|---|
| Normal year change information (From 2014 to 2021) | 0.4600 | <10% | 38 | Stable | |
| | 1.2100 | 10–30% | 29 | Unstable | See Figure 8a |
| | 1.9500 | >30% | 4 | Instability | |
| Abnormal year change information (From 2014 to 2021) | 0.0700 | <10% | 0 | Stable | |
| | 0.9500 | 10–30% | 8 | Unstable | See Figure 8b |
| | 8.9100 | >30% | 63 | Instability | |
| Simulation of normal year change information (From 2022 to 2025) | 0.0046 | <10% | -- | Stable | |
| | 0.0107 | 10–30% | -- | Unstable | See Figure 9a |
| | 0.0045 | >30% | -- | Instability | |
| Simulation of abnormal year change information (From 2022 to 2025) | 0.0360 | <10% | -- | Stable | |
| | 1.3080 | 10–30% | -- | Unstable | See Figure 9b |
| | 3.5230 | >30% | -- | Instability | |

According to the relevant standards of gravity erosion classification, the spatial distributions of the slope stability under the two environmental conditions from 2014 to 2021 were plotted (Figure 8a,b). Similarly, the simulation results for the next five years were mapped (Figure 9a,b).

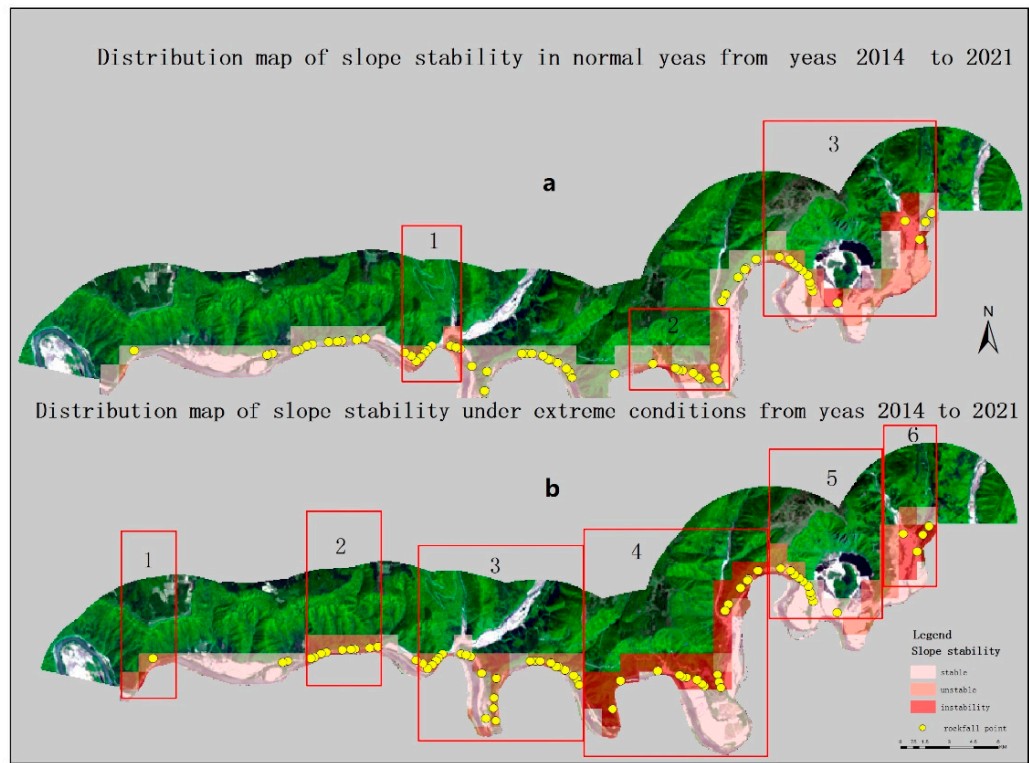

**Figure 8.** The schematic diagrams of slope stability under two natural environmental conditions from 2014 to 2021. (**a**,**b**) respectively for the explanation of the paper.

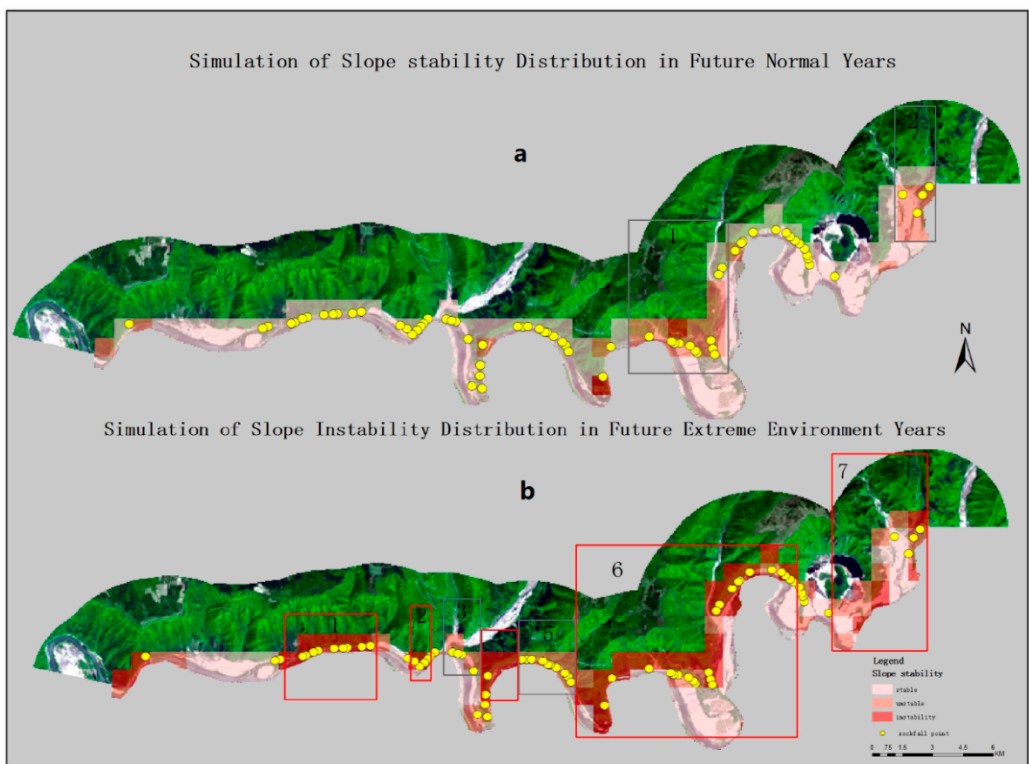

**Figure 9.** The schematic diagrams of the simulation of slope stability under two natural environmental conditions in the next five years. (**a**,**b**) respectively for the explanation of the paper.

### 5.2. Field Investigation

In combination with the simulated RS results, a field survey was carried out on 71 highway slopes in the study area, 9 of which were found to be inconsistent with the actual situation. The simulation results showed that the area of iron staining abnormalities was small, but in the actual survey, it was found that there were rockfills on the slopes. The survey indicated that this was due to accumulation at the foot of the slope caused by the fall of small local gravel, and the slope surface was relatively complete as a whole. Under the influence of heavy rainfall and other factors, local rockfall occurred, which failed to produce a large range of iron film peeling in the grid area. In addition, a large rockfill (field investigation no. 68) with a volume of 11.88 m$^3$ was identified. This was caused by a previous rock mass collapse, which was not in the time series of this analysis. Based on the field investigation, the accuracy rate was determined to be 87.32%, as shown in Table 5.

**Table 5.** The comparison of the changes of field-measured and indoor iron staining anomalies.

| Field Investigation No. | Zone ID | Field-Measured Data of Slope Rock and Soil Mass (Unit: m) | | | | | | | Volume of Slope Toe Deposits (m$^3$) | $I_{TRYC}$ |
| | | Slope Top Elevation (m) | Footing Elevation (m) | Slope Length (m) | Slope Width | Slope Height (m) | Depth of Completely Weathered Zone (m) | Unloading Crack Depth (m) | | |
|---|---|---|---|---|---|---|---|---|---|---|
| 1 | 1 | 582.60 | 561.60 | 31.00 | 130.00 | 22.00 | 1.20 | 0.00 | 2.60 | 0.887 |
| 2 | | 555.60 | 530.60 | 33.00 | 249.00 | 25.00 | 1.10 | 0.60 | 10.00 | 0.951 |
| 3 | | 558.50 | 541.00 | 17.50 | 44.00 | 11.00 | 0.00 | 0.00 | 12.00 | 0.951 |
| 4 | | 579.40 | 545.40 | 36.00 | 283.00 | 34.00 | 1.80 | 0.00 | 2.00 | 0.774 |
| 5 | | 560.20 | 534.20 | 30.00 | 191.00 | 26.00 | 0.80 | 0.60 | 1.50 | 0.737 |
| 6 | 2 | 577.30 | 539.30 | 38.00 | 254.00 | 22.00 | 1.20 | 0.00 | 1.40 | 0.737 |
| 7 | | 568.80 | 548.30 | 24.00 | 89.00 | 20.50 | 1.60 | 0.00 | 2.34 | 0.991 |
| 8 | | 565.30 | 554.30 | 13.00 | 79.00 | 11.00 | 1.50 | 0.80 | 1.50 | 0.991 |
| 9 | | 564.20 | 547.20 | 20.00 | 79.00 | 17.00 | 1.20 | 0.00 | 4.50 | 0.903 |
| 10 | | 550.40 | 543.40 | 6.00 | 252.00 | 7.00 | 0.00 | 0.00 | 5.10 | 0.903 |
| 11 | | 574.90 | 573.10 | 3.00 | 177.00 | 1.80 | 1.60 | 0.00 | 3.75 | 0.490 |
| 12 | | 600.30 | 589.30 | 19.00 | 169.00 | 11.00 | 0.00 | 0.00 | 7.00 | 0.921 |
| 13 | | 581.60 | 567.60 | 9.00 | 160.00 | 6.50 | 0.00 | 0.00 | 3.00 | 0.551 |
| 14 | | 582.00 | 574.00 | 10.00 | 351.00 | 8.00 | 1.60 | 0.00 | 3.15 | 0.551 |
| 15 | | 585.60 | 552.60 | 43.00 | 12.00 | 33.00 | 0.00 | 0.00 | 2.20 | 0.374 |
| 16 | | 557.20 | 551.20 | 7.00 | 65.00 | 6.00 | 0.00 | 0.00 | 18.75 | 0.909 |
| 17 | | 587.50 | 571.50 | 25.00 | 235.00 | 16.00 | 1.20 | 0.00 | 3.63 | 0.372 |
| 18 | | 578.90 | 568.90 | 14.00 | 212.00 | 10.00 | 0.80 | 0.00 | 1.20 | 0.372 |
| 19 | | 592.10 | 573.10 | 31.00 | 203.00 | 19.00 | 1.20 | 0.00 | 3.60 | 0.372 |
| 20 | | 563.40 | 556.00 | 10.00 | 40.00 | 7.40 | 1.80 | 0.00 | 3.00 | 0.281 |
| 21 | | 570.30 | 553.20 | 26.00 | 133.00 | 17.00 | 1.10 | 0.80 | 3.75 | 0.541 |
| 22 | 3 | 556.00 | 548.00 | 11.00 | 77.00 | 8.00 | 1.00 | 0.70 | 3.15 | 0.541 |
| 23 | | 570.80 | 546.80 | 24.00 | 64.00 | 24.00 | 1.10 | 0.70 | 1.20 | 0.541 |
| 24 | | 569.10 | 543.10 | 28.00 | 50.00 | 26.00 | 0.80 | 0.00 | 8.00 | 0.569 |
| 25 | | 558.00 | 542.00 | 19.00 | 127.00 | 16.00 | 0.80 | 0.00 | 2.64 | 0.569 |
| 26 | | 567.40 | 548.40 | 22.00 | 30.00 | 19.00 | 2.00 | 0.70 | 3.00 | 0.569 |
| 27 | | 563.40 | 545.40 | 26.00 | 56.00 | 18.00 | 1.80 | 0.00 | 9.00 | 0.383 |
| 28 | | 568.70 | 546.70 | 22.00 | 74.00 | 19.00 | 0.80 | 0.00 | 3.00 | 0.383 |
| 29 | | 567.00 | 561.00 | 8.00 | 169.00 | 6.00 | 0.80 | 0.00 | 9.00 | 0.884 |
| 30 | | 575.40 | 560.40 | 17.00 | 17.00 | 15.00 | 0.80 | 0.00 | 1.00 | 0.884 |
| 31 | | 595.10 | 560.10 | 40.00 | 258.00 | 35.00 | 0.80 | 0.00 | 2.00 | 0.884 |
| 32 | | 594.90 | 651.90 | 38.00 | 99.00 | 33.00 | 0.80 | 0.00 | 1.20 | 0.581 |
| 33 | | 606.00 | 562.00 | 50.00 | 151.00 | 44.00 | 1.50 | 1.10 | 1.88 | 0.581 |
| 34 | | 569.90 | 561.90 | 14.00 | 94.00 | 8.00 | 1.50 | 0.80 | 3.00 | 0.696 |

**Table 5.** *Cont.*

| Field Investigation No. | Zone ID | Field-Measured Data of Slope Rock and Soil Mass (Unit: m) | | | | | | | | $I_{TRYC}$ |
|---|---|---|---|---|---|---|---|---|---|---|
| | | Slope Top Elevation (m) | Footing Elevation (m) | Slope Length (m) | Slope Width | Slope Height (m) | Depth of Completely Weathered Zone (m) | Unloading Crack Depth (m) | Volume of Slope Toe Deposits (m³) | |
| 35 | | 581.60 | 567.60 | 17.00 | 171.00 | 14.00 | 0.60 | 0.30 | 5.25 | 0.797 |
| 36 | | 597.70 | 583.70 | 21.00 | 108.00 | 14.00 | 1.10 | 0.70 | 1.12 | 0.400 |
| 37 | | 587.10 | 568.10 | 29.00 | 256.00 | 19.00 | 1.00 | 0.80 | 6.00 | 0.800 |
| 38 | | 588.00 | 576.00 | 14.00 | 21.00 | 12.00 | 1.00 | 0.80 | 2.66 | 0.800 |
| 39 | | 587.10 | 583.10 | 6.00 | 40.00 | 4.00 | 1.00 | 0.70 | 15.00 | 0.800 |
| 40 | | 592.60 | 580.60 | 18.00 | 151.00 | 12.00 | 1.50 | 0.80 | 5.25 | 0.800 |
| 41 | | 612.40 | 597.40 | 21.00 | 97.00 | 15.00 | 1.00 | 0.80 | 9.00 | 0.800 |
| 42 | | 617.60 | 595.60 | 30.00 | 11.00 | 22.00 | 1.50 | 1.10 | 0.45 | 0.800 |
| 43 | 4 | 649.70 | 634.70 | 21.00 | 162.00 | 15.00 | 0.00 | 0.00 | 9.75 | 0.800 |
| 44 | | 660.90 | 635.90 | 32.00 | 167.00 | 25.00 | 0.80 | 0.00 | 1.08 | 0.800 |
| 45 | | 669.50 | 644.50 | 33.00 | 90.00 | 25.00 | 0.80 | 0.00 | 2.25 | 0.800 |
| 46 | | 589.60 | 577.60 | 17.00 | 212.00 | 12.00 | 0.80 | 0.00 | 12.00 | 0.758 |
| 47 | | 633.50 | 627.00 | 8.00 | 102.00 | 6.50 | 1.20 | 0.80 | 3.63 | 0.450 |
| 48 | | 616.00 | 610.00 | 10.00 | 305.00 | 6.00 | 0.80 | 0.60 | 3.00 | 0.450 |
| 49 | | 626.50 | 620.00 | 7.00 | 94.00 | 6.50 | 1.20 | 0.80 | 1.20 | 0.544 |
| 50 | | 641.00 | 634.00 | 10.00 | 245.00 | 7.00 | 1.10 | 0.70 | 3.60 | 0.544 |
| 51 | | 665.00 | 630.00 | 42.00 | 115.00 | 35.00 | 0.00 | 0.00 | 1.88 | 0.709 |
| 52 | | 478.40 | 460.40 | 22.00 | 402.00 | 18.00 | 1.20 | 0.00 | 3.00 | 0.673 |
| 53 | | 643.60 | 627.60 | 22.00 | 185.00 | 16.00 | 0.80 | 0.00 | 15.00 | 0.122 |
| 54 | | 695.70 | 665.70 | 37.00 | 107.00 | 30.00 | 0.00 | 0.00 | 1.10 | 0.855 |
| 55 | 5 | 780.10 | 660.10 | 177.00 | 270.00 | 120.00 | 1.20 | 0.00 | 5.25 | 0.579 |
| 56 | | 651.00 | 626.00 | 30.00 | 205.00 | 25.00 | 0.20 | 0.30 | 0.90 | 0.579 |
| 57 | | 640.20 | 626.70 | 15.00 | 179.00 | 13.50 | 1.20 | 0.00 | 1.50 | 0.579 |
| 58 | | 649.50 | 645.00 | 7.00 | 151.00 | 4.50 | 1.20 | 0.00 | 1.12 | 0.800 |
| 59 | | 648.90 | 623.00 | 35.00 | 52.00 | 27.00 | 1.20 | 0.00 | 6.00 | 0.800 |
| 60 | 6 | 624.50 | 619.00 | 7.00 | 10.00 | 5.50 | 1.20 | 0.00 | 2.66 | 0.800 |
| 61 | | 639.00 | 619.00 | 44.00 | 227.00 | 20.00 | 0.00 | 0.00 | 1.50 | 0.800 |
| 62 | | 643.20 | 639.20 | 5.00 | 172.00 | 4.00 | 1.60 | 0.00 | 10.00 | 0.800 |
| 63 | | 562.80 | 544.80 | 22.00 | 151.00 | 18.00 | 0.00 | 0.00 | 0.60 | 0.200 |
| 64 | | 664.60 | 640.60 | 31.00 | 100.00 | 24.00 | 0.00 | 0.00 | 0.85 | 0.200 |
| 65 | | 573.00 | 557.00 | 22.00 | 52.00 | 16.00 | 0.00 | 0.00 | 0.40 | 0.383 |
| 66 | | 568.70 | 554.70 | 23.00 | 233.00 | 14.00 | 0.00 | 0.00 | 0.48 | 0.581 |
| 67 | -- | 609.70 | 605.70 | 5.00 | 73.00 | 4.00 | 1.20 | 0.00 | 5.25 | 0.419 |
| 68 | | 608.80 | 591.80 | 23.00 | 129.00 | 17.00 | 0.00 | 0.00 | 11.88 | 0.127 |
| 69 | | 632.20 | 619.20 | 19.00 | 27.00 | 13.00 | 1.20 | 0.00 | 0.45 | 0.217 |
| 70 | | 634.00 | 620.00 | 18.00 | 161.00 | 12.00 | 0.00 | 0.00 | 0.45 | 0.217 |
| 71 | | 637.60 | 631.10 | 11.00 | 176.00 | 6.50 | 1.20 | 0.00 | 9.00 | 0.217 |

Table notes: ① The Zone ID corresponds to Figure 8b. ② The volume of deposits at the foot of the slope is an important indicator of slope stability in gravity-based geological disasters such as collapse or rockfall. In general, the more deposits at the toe of the slope, the more unstable the slope. ③ See Equation (11) for the calculation method of $I_{TRYC}$.

### 5.3. Discussion

(1) The S3K highway slope collapse disaster point is very concentrated, which, from the field survey perspective, was mainly due to the rock weathering of the high and steep slope in the natural environment, resulting in a reduction in the strength of the potential structural plane of the rock mass. This caused rock damage, which, under the action of environmental forces, ultimately gradually formed the current collapse point. It was also found that the slope deposits are mainly located at the toe of the rockfall disaster points. Via the experimental simulation and historical extreme weather (heavy rainfall) data, it can be determined that the stability of the slope in this area is closely related to the change in the meteorological conditions, and there exists periodicity (see Figure 8).

(2) According to the survey data (see Table 5), the statistical analysis indicates that the degree of correlation between the iron staining anomaly value ($Y_{TRYC}$) and the slope top elevation ($X_1$), the slope toe elevation ($X_2$), the fully weathered depth ($X_3$), and

the unloading crack depth ($X_4$) is 0.93; the $R^2$ value is 0.87; and the relationship is $Y_{TRYC} = 0.000905X_1 + 0.000062X_2 + 0.0251X_3 + 0.0873X_4$. Table 4 reveals that the abnormal change area of iron staining anomalies was found to be positively correlated with the amount of slope deposits. The instability of the slope in the area was found to be the most intense in the central region (Figure 8(a2)), and there were obvious differences between the western and eastern areas of the central region. Through field investigation, it was found that the weathering of the sloping rock mass in the central section is extremely serious, and the maximum thickness reaches 7.4 m. A trenching project was carried out in a typical small area, and the intact rock mass was not seen at 3.5 m. Moreover, the bottom of the slope was in a fully differentiated state. The debris at the slope toe accounted for 40%, and the soil accounted for more than 50%.

To the east of the central section, the rock mass is moderately to strongly weathered, and the deposits at the slope toe are mainly gravels. To the west of the central section, the rock is mainly moderately weathered, and it is hard and brittle. The overall massive rock mass is found at about 50 to 200 cm below the bottom of the slope in this area, and the accumulation at the foot of the slope is reduced. Some of the main data obtained from the survey are reported in Table 6.

**Table 6.** The field investigation characteristics of the section S3K highway slope.

| S3K Partition | Longitude and Latitude Coordinates | Slope Geometry | | | Volume Interval of Deposits at the Slope Toe (m³) | Rock Mass Structure |
|---|---|---|---|---|---|---|
| | | Length (m) | Width (m) | Height (m) | | |
| Central section | 127°50′26.8″,41°25′16.4″ ---127°51′44.1″, 41°26′10″ | 71–114 | 11–28 | 49–83 | 2.2–18.75 | The rock mass of the slope is broken overall, and the bottom of the slope is weathered completely. |
| East of the central section | 127°55′26.70″, 41°27′4.50″--- 127°55′39.5″, 41°27′27.0″ | 20–64 | 17–25 | 10–40 | 0.4–9.0 | |
| West of the central section | 127°45′46″, 41°25′23″--- 127°46′16.80″, 41°25′29.10″ | 10–44 | 6–30 | 10–50 | 0.2–3.7 | The rock mass structure of the slope is mainly expressed as a whole block. |

(3) Based on the field investigation, the accumulation of the slope toe in the central area is 122.21 m³, accounting for 42.27% of the total accumulation. The slope toe accumulation of area 3 (Figure 8a) is 18.23 m³, accounting for 6.31% of the total accumulation. Based on the statistics of abnormal iron staining data under abnormal historical natural environmental conditions (Figure 8b), the extremely unstable areas of the slope are concentrated in areas 3 and 5, and the measured accumulation at the foot of the slope is 89.7 m³, accounting for 31.14% of the total accumulation in the whole area. The unstable state is predominantly distributed in areas 1, 2, and 4, and the accumulation at the slope toe is 190.6 m³, accounting for 66.08% of the total area. The slope stability under an extreme environment in the future was simulated, and the results reveal many unstable slopes (as shown in Figure 9(b1,b2,b4,b6,b7)). For normal years in the future, it was found that the overall slope is stable except for the unstable focal region (as shown in Figure 9(a1,a2)).

In summary, the focus of disaster prevention in this area should be on the inner section. Under the influence of rainfall, the debris on the surface of the slope will migrate with the scouring of rainwater. The instability of a high and steep slope in a basalt area can be sufficiently analyzed via iron staining anomalies. Moreover, the results are consistent with previous research results [33].

## 6. Conclusions

First, it was considered that the more developed the fractures in a high and steep highway slope rock mass, the stronger the water action. Moreover, the higher degree of development of a thin iron film on the corresponding fracture surface in basalt areas with high iron content leads to the varying occurrence of iron staining anomalies in the rock mass. Iron staining anomalies are a type of rock mass alteration that have a close causal relationship with the weathering, crushing, and mechanical properties of a rock mass, and the material migration of the surface debris attached to the rock mass occurs under the joint action of gravity, rainfall, and other factors. This affects the change in iron staining anomalies, which is easy to analyze from RS images.

The slope of section S3K of a highway in Changbai County, China, a typical volcanic rock region, was taken as an example, and the relationship between the iron staining anomalies and rock mass fragmentation was determined via field investigation (see Table A1). Then, the ANN-CA model was established, and the control factors suitable for this area were selected. The future slope instability in normal and extreme years was then simulated. It is important to note that iron staining anomalies can effectively reflect the migration of weathered debris on the surface of the slope in this area.

Finally, a useful conclusion was drawn; namely, that it is feasible to reinterpret the alteration information in the prospecting field from the perspective of geological engineering, and iron staining anomalies can be used as an indicator to study the stability of slopes in basalt areas. Iron staining anomalies have a definite expansion effect on the application of geoscience RS. This study is valuable and can provide a reliable reference for relevant research by other scholars.

**Author Contributions:** Methodology, funding acquisition, supervision, project administration, S.Z.; Formal analysis, investigation, data curation, writing original draft preparation, L.Q., H.M., L.S. and X.W. All authors have read and agreed to the published version of the manuscript.

**Funding:** This research was funded by the Science & Technology Fundamental Resources Investigation Program (Grant No. 2022FY100701), National Natural Science Foundation of China (NSFC) (No. 41971151), Key Joint Program of National Natural Science Foundation of China (NSFC), and Heilongjiang Province for Regional Development (No. U20A2082).

**Data Availability Statement:** Not applicable.

**Acknowledgments:** I would like to show my deepest gratitude to my supervisor, Shuying Zang, a respectable, responsible tutor, who has provided me with valuable guidance in every stage of the writing of this thesis. Without her enlightening instruction, impressive kindness, and patience, I could not have completed my paper. At the same time, I thank my collaborators Haoran Man, Li Sun and Xiangwen Wu.

**Conflicts of Interest:** We declare that we have no conflict of interest.

## Appendix A

**Table A1.** The field investigation and interpretation comparison.

| No. | Microtopography | Slope Height (m) | Slope Width (m) | Slope Length (m) | Field Investigation of the Rock Mass Structure | Remote Sensing Analysis Results |
|---|---|---|---|---|---|---|
| 0 | Steep slope | 20 | 227 | 44 | Integral block | Bulk |
| 1 | Steep cliffs | 17 | 129 | 23 | Block structure | Block structure |
| 2 | Steep cliffs | 14 | 179 | 15 | Block structure | Block structure |
| 3 | Steep cliffs | 25 | 205 | 30 | Integral block | Integral block |

**Table A1.** *Cont.*

| No. | Microtopography | Slope Height (m) | Slope Width (m) | Slope Length (m) | Field Investigation of the Rock Mass Structure | Remote Sensing Analysis Results |
|---|---|---|---|---|---|---|
| 4 | Steep slope | 120 | 270 | 177 | Block structure | Block structure |
| 5 | Steep cliffs | 30 | 107 | 37 | Block structure | Block structure |
| 6 | Steep cliffs | 35 | 115 | 42 | Integral block | granular structure |
| 7 | Gentle slope | 7 | 245 | 10 | Integral block | granular structure |
| 8 | Steep slope | 15 | 97 | 21 | Block structure | Block structure |
| 9 | Steep slope | 12 | 151 | 18 | Block structure | Block structure |
| 10 | Steep slope | 12 | 21 | 14 | Fragmentation structure | Fragmentation structure |
| 11 | Steep cliffs | 19 | 256 | 29 | Block structure | Block structure |
| 12 | Steep slope | 12 | 212 | 17 | granular structure | granular structure |
| 13 | Steep cliffs | 14 | 171 | 17 | Block structure | Block structure |
| 14 | Steep slope | 16 | 235 | 25 | Block structure | Block structure |
| 15 | Steep slope | 8 | 94 | 14 | Block structure | Block structure |
| 16 | Steep cliffs | 44 | 151 | 50 | Block structure | Block structure |
| 17 | Steep slope | 15 | 17 | 17 | Integral block | granular structure |
| 18 | Steep slope | 6 | 169 | 8 | Block structure | Block structure |
| 19 | Steep slope | 8 | 351 | 10 | Block structure | Block structure |
| 20 | Steep slope | 18 | 56 | 26 | Block structure | Block structure |
| 21 | Steep slope | 7 | 40 | 10 | granular structure | granular structure |
| 22 | Steep cliffs | 17 | 79 | 20 | Integral block | Integral block |
| 23 | Steep slope | 24 | 100 | 31 | Block structure | Block structure |
| 24 | Steep slope | 18 | 151 | 22 | Block structure | Block structure |
| 25 | Steep slope | 22 | 130 | 31 | Block structure | Block structure |
| 26 | Steep slope | 33 | 99 | 38 | Block structure | Block structure |
| 27 | Steep slope | 14 | 233 | 23 | Fragmentation structure | Fragmentation structure |
| 28 | Steep slope | 35 | 258 | 40 | Integral block | granular structure |
| 29 | Steep slope | 11 | 169 | 19 | Block structure | Block structure |
| 30 | Steep slope | 19 | 74 | 22 | Block structure | Block structure |
| 31 | Steep slope | 16 | 52 | 22 | Block structure | Block structure |
| 32 | Steep cliffs | 7 | 252 | 6 | Integral block | granular structure |
| 33 | Steep cliffs | 11 | 79 | 13 | Block structure | Block structure |
| 34 | Steep cliffs | 21 | 89 | 24 | Block structure | Block structure |
| 35 | Steep cliffs | 19 | 203 | 31 | Block structure | Block structure |
| 36 | Steep slope | 27 | 52 | 35 | granular structure | granular structure |
| 37 | Steep slope | 6 | 10 | 7 | Integral block | Integral block |
| 38 | Steep slope | 4 | 172 | 5 | Integral block | Integral block |
| 39 | Steep slope | 5 | 151 | 7 | Fragmentation structure | Fragmentation structure |
| 40 | Steep slope | 4 | 73 | 5 | Block structure | Block structure |

**Table A1.** *Cont.*

| No. | Microtopography | Slope Height (m) | Slope Width (m) | Slope Length (m) | Field Investigation of the Rock Mass Structure | Remote Sensing Analysis Results |
|-----|-----------------|------------------|-----------------|------------------|------------------------------------------------|--------------------------------|
| 41 | Steep slope | 7 | 176 | 11 | Block structure | Block structure |
| 42 | Gentle slope | 12 | 161 | 18 | Block structure | Block structure |
| 43 | Steep slope | 13 | 27 | 19 | Block structure | Block structure |
| 44 | Steep slope | 16 | 185 | 22 | Block structure | Block structure |
| 45 | Steep slope | 18 | 402 | 22 | Block structure | granular structure |
| 46 | Steep cliffs | 7 | 94 | 7 | Block structure | Block structure |
| 47 | Steep slope | 5 | 30 | 8 | granular structure | granular structure |
| 48 | Steep slope | 6 | 305 | 10 | Fragmentation structure | Fragmentation structure |
| 49 | Steep slope | 15 | 162 | 21 | Block structure | Block structure |
| 50 | Steep slope | 25 | 167 | 32 | Block structure | Block structure |
| 51 | Steep slope | 25 | 90 | 33 | Block structure | Block structure |
| 52 | Steep slope | 22 | 11 | 30 | Block structure | Block structure |
| 53 | Steep slope | 4 | 40 | 6 | Integral block | granular structure |
| 54 | Steep slope | 14 | 108 | 21 | Fragmentation structure | Fragmentation structure |
| 55 | Steep slope | 10 | 212 | 14 | Integral block | Integral block |
| 56 | Steep cliffs | 6 | 65 | 7 | Block structure | Block structure |
| 57 | Steep slope | 7 | 160 | 9 | Integral block | Integral block |
| 58 | Steep slope | 2 | 177 | 3 | Fragmentation structure | Fragmentation structure |
| 59 | Steep slope | 33 | 12 | 43 | Block structure | Block structure |
| 60 | Steep slope | 19 | 30 | 22 | Integral block | Integral block |
| 61 | Steep slope | 26 | 50 | 28 | Integral block | Integral block |
| 62 | Steep slope | 16 | 127 | 19 | Integral block | Integral block |
| 63 | Steep slope | 24 | 64 | 24 | Integral block | Integral block |
| 64 | Steep slope | 8 | 77 | 11 | Integral block | Integral block |
| 65 | Steep slope | 17 | 133 | 26 | granular structure | granular structure |
| 66 | Steep slope | 22 | 254 | 38 | Integral block | Integral block |
| 67 | Steep slope | 26 | 191 | 30 | Block structure | Block structure |
| 68 | Steep slope | 34 | 283 | 36 | Block structure | Block structure |
| 69 | Steep cliffs | 25 | 249 | 33 | Integral block | Integral block |
| 70 | Steep cliffs | 11 | 44 | 18 | Block structure | Block structure |

Table notes: ① The longitude and latitude coordinate data are classified and not listed in the table. ② The accuracy rate is 88.57%, and there are eight inaccuracies. This is mainly because the slope was in a stable state for a long period of time, which increased the iron film on the slope surface in the natural environment. This displayed a strong result in the RS images.

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
