# Peer review of "Determination of the Stability of a High and Steep Highway Slope in a Basalt Area Based on Iron Staining Anomalies"

_remotesensing, doi:10.3390/rs15123021_

Round 1
Reviewer 1 Report (New Reviewer)
This is an interesting paper and I quite like the linkage between the evidence of basalt alteration from Landsat and the slope condition. I confess to becoming lost in some of the discussion and analytical notation and suggest that you to find ways of simplifying the text. Most readers for example are familiar with ANN and, in my opinion you spend too much time describing and illustrating the technique. I have attached some comments on the downloaded pdf that you might find useful

Author Response
Since last year until now, this paper has undergone different reviewers. I am very grateful for the repeated opinions. This is the fifth adjustment, improvement, and necessary modification.
Reviewer 2 Report (New Reviewer)
The author have tried to evaluate and determine the evolution of the stability of high 14 and steep highway slopes in basalt areas. Follwing are the comments:
1. English should be checked by a professional as there are many grammatical errors.
2. Abstract is not written well and should be rewritten to include quantitative findings.
3. Author should highlight the importance of use of ANN-CA over other modelling approahes
4. What is the novelty of the approach used, and how it is different from other methods used worldwide
5. Pease validate the obtained results with field condition
6. Please highlight the innovative component of the work
7. Some of the relevant references should be added
Author Response
Since last year until now, this paper has undergone different reviewers. I am very grateful for the repeated opinions. This is the fifth adjustment, improvement, and necessary modification. 1. English should be checked by a professional as there are many grammatical errors. 【Solution】checked 2 Abstract is not written well and should be rewritten to include quantitative findings. 【Solution】Modify slightly. Innovative and valuable findings are clearly described in the abstract. That is, the iron staining anomaly can be used to identify the stability of the slope in the basalt area. 3 uthor should highlight the importance of use of ANN-CA over other modelling approahes ,and What is the novelty of the approach used, and how it is different from other methods used worldwide. 【Solution】Summative descriptions and citations added. 4. Pease validate the obtained results with field condition. 【Response】The fifth section introduces it very clearly 5. Please highlight the innovative component of the work. 【Solution】The abstract has been revised and highlighted in the conclusion section. The use of iron staining anomalies to determine the stability of slopes in basalt areas is an attempt and innovation. 6. Some of the relevant references should be added. 【Solution】added.
Reviewer 3 Report (New Reviewer)
The paper deals with theoretical and practical approaches for the assessment of potential risks for the stability of steep rock slopes along higways.
Some points to improve and complete the draft.
1) Avoid the long extension of the title (Taking the S3K section of a highway in Changbai County, China, as an example)- and simply put at the end of it: the case of Changbai County, China.
2) RS not suitable as keyword
3) Line 153 correct word neuon
4) For fig. 3.3 the criterion of iron staining presence is not sufficiently emphasized
5) Equations 3-2 to 3-4 should be described also in a non technical way, ir order to understand their meaning for the procedure, and also 5-1 eq.
6) Line 284: comment the cause for these anomalies (geology, water, weathering etc) also referred to metric scale of the extension of these anomalies
7) Line 484: comment extensively “resulting in a reduction in the strength of the rock mass potential structural plane” as geostructural elemt for triggering the landslide
8) In Conclusion provide also a comment for higway management (meteo surveillance, topographical targets, provisional reinforcement systems …)
9) I suggest to add three references that are meaningful for introduction section. The 3 references are :
9.1) SLOPE STABILITY USING REMOTE SENSING AND GEOGRAPHIC INFORMATION SYSTEM ALONG KARAK HIGHWAY, MALAYSIA, February 2010, Thesis for: MSc Remote, by
· Hamdan Omar,
open access online.
9.2) Remote sensing GIS-based landslide susceptibility & risk modeling in Darjeeling–Sikkim Himalaya together with FEM-based slope stability analysis of the terrain
· Sankar Kumar Nath,
· Arnab Sengupta &
· Anand Srivastava
Natural Hazards volume 108, pages3271–3304 (2021),,open access
9.3) Reinforcement design and control of rock slopes above tunnel portals in northern Italy
Del Greco, O., Oggeri, C.
International Journal of Rock Mechanics and Mining Sciences, DOI
10.1016/j.ijrmms.2004.03.136
REGARDS
Author Response
Since last year until now, this paper has undergone different reviewers. I am very grateful for the repeated opinions. This is the fifth adjustment, improvement, and necessary modification. 1 Avoid the long extension of the title (Taking the S3K section of a highway in Changbai County, China, as an example)- and simply put at the end of it: the case of Changbai County, China. 【Solution】deleted, and reflected in the abstract of the paper. 2 RS not suitable as keyword 【Solution】 Modified. 3 Line 153 correct word neuon 【Solution】 make corrections,thank you. 4 For fig. 3.3 the criterion of iron staining presence is not sufficiently emphasized 【Response】 This is the best problem I have ever seen. Iron staining anomalies are usually applied in the field of mineral exploration. I also found this problem during my research process. After consulting some data, I did not find any classification standards, but mostly used the Interactive Stretching function module in remote sensing software to perform interactive stretching to highlight the information of iron staining anomalies. This paper reinterprets it from an engineering perspective and believes that rock fractures the more developed the joints, the stronger the iron staining anomaly, which has been confirmed in field investigations (see article Attached table 1) 5 Equations 3-2 to 3-4 should be described also in a non technical way, ir order to understand their meaning for the procedure, and also 5-1 eq. 【Response】By re-examining the text description, the meaning is clear and the relevant paper is also cited. I think this is a paper, there is no need to describe the text like the report. 6. Line 284: comment the cause for these anomalies (geology, water, weathering etc) also referred to metric scale of the extension of these anomalies 【Response】The results of the actual investigation are shown in Attached table 1 and Section. 7. Line 484: comment extensively “resulting in a reduction in the strength of the rock mass potential structural plane” as geostructural elemt for triggering the landslide. 【Response】Refer to the local geological disaster survey data, and the field survey of the relevant text, the description is reasonable. Reference 33 also illustrates this point. 8 In Conclusion provide also a comment for higway management (meteo surveillance, topographical targets, provisional reinforcement systems …) 【Solution】Delete inappropriate statements. 9 I suggest to add three references that are meaningful for introduction section. The 3 references are :……. 【Solution】added.
This manuscript is a resubmission of an earlier submission. The following is a list of the peer review reports and author responses from that submission.
Round 1
Reviewer 1 Report
This is a research on the idea that remote sensing of iron staining anomalies on slope stability. Idea is good, and readers may have an interest on it. However, the details are not well explained.
First, based on the conclusions, it is written as "the relationship between the iron stain anomalies and rock mass fragmentation was established via field investigation.". However, in section 5.2 Field investigation, there is no established relationships are shown: only Table5-2 and complicated explanations. What is the definition of "ratio of iron staining anomalies changes to grid area"?... what is the "grid" here??? Is the rock mass fragmentation equals to the "volume of slope toe deposit"???? The reviewer cannot follow the details.
Second, based on the conclusions, it is written as "the ANN-CA model was established". However, how ANN and CA are united is not well explained. The reviewer can find "Establishing ANN-CA model" in Fig.3, however, there are no explanations.
By the way, why the target area in Fig.1 has a strange shape. The upper lines are quite straight, but the lower lines are curved. How this area related to the S3K section of highway??? The reviewer cannot understand the overall framework of the study.
Reviewer 2 Report
The paper by LiHui Qian, ShuYing Zang, HaoRan Man, Li Sun and XiangWen Wu presents a method based on ANN-CA to determine stability of highway slopes in a basalt area using iron staining images. The manuscript lacks some parts like a detailed definition of the used method (i.e. ANN-CA). In addition, it is not clear how the authors validate the results of the proposed approach; the linear regression and R squared values presented in lines 414-415 is a rather simple method for validating a complex method such as ANN-CA. Finally, de document lacks clarity in some parts, English needs to be improved. Some comments are listed below:
1. Lines 79-85: meaning not clear. Rewrite
2. Line 165: it would be helpful to describe the Moore neighborhood here.
3. The ANN is not explained at all, what kind of problem is being solved? Supervised learning? In that case, what is the loss function? What is the relation of CA with ANN? A complete description of ANN-CA is needed before jumping to section 4. This description should allow reproducibility and ideally authors should share their code.
4. In line 188: what does it mean that data were registered uniformly?
5. Line 192: reference 27 is in Chinese, therefore you should briefly explain what are bands 2, 4 ,5, and 6 and why are they selected.
6. Why is PC1 selected for analysis, what is the criteria?
7. Lines 199-201: a more in detail explanation and meaning of the mean digital number and its calculation is needed. The meaning of “According to the mathematical meaning of the probability density distribution curve” is not clear.
8. Legend in figure 4-1 illegible, image with poor resolution as well.
9. Link in line 227 shows September weather, you can place it after mentioning typhoons Bawei and Meissack.
10. Lines 245-246: Why the instability of the slope is manifested as the exchange of energy?
11. Line 250: what is the architecture of the ANN? Activation functions, connection topology, etc.
12. In equation 4-1, if you convert from degrees to radians as it seems you do, then you should describe the “slope” variable as being in degrees, otherwise just omit the conversion in the equation.
13. Paragraph after equation 4-3: it is not clear what is the value exceeding 7 in line 271, if Yang [50] suggested that the maximum expected LAI value is 3, how can it be 7 in the present case?
14. In equation 4-6, better change “ann” subscript by ANN to avoid confusion with “annual”. What does it mean that 1+(-ln gamma)^alpha is a random factor? Define gamma and alpha.
15. In lines 299-300, what is the meaning of “In the model, the integrity of the rock and soil was set as a sign of stability, so that the reference type when the computer terminates the condition. 301” ?
16. In the article it is mentioned that the taller the slope the more likely to be unstable, which is a logic relation. Meanwhile, in the relation Y=f(X1,X2,X3) in line 414, instead of the slope height, X1 and X2 are used, then it would be expected that X2 had a negative coefficient because height= X1-X2. How do you explain such result?
Specific comments
Line 74: water infiltrates rather than is immersed
Line 76-79, rephrase
3à0 in line 159 seems to be repeated, 2à1 instead?
Figures and table change numbering criteria throughout the document, they need to be unified.
Line 236: there is no section 1.1 in the document.
Round 2
Reviewer 2 Report
A comment on a few remarks is given below:
Remark 2: Contrary to what the authors state in the reply letter, Moore neighborhood remains without being described.
Remark 3: There is no section 2.3, did you mean figure?
Data normalization is a quite standard procedure in machine learning; it is not needed to add equation 2-1. The newly added section 3.3 mainly describes the functioning of a neural network, but does not clearly demonstrate how the link ANN-CA is implemented. It seems that the only allusion to that link is made via equation 2-2, but again, this is not enough. Is “k” in equation 2-2 capital K in the definition? Again, in figure 2-5, the transition between initial stage and input layer “x_i” is not explained, I guess that the CA is employed here, but it is not depicted in the figure nor in the diagram of figure 2-4. The same for output layer/final stage.
Remark 11: Section 2.3 does not exist. In section 3.3 some aspects of the neural network are defined. It seems that it consists in a vanilla neural network with three layers. Some parameters remain unrevealed, e.g., number of neurons per layer, activation functions, type of regularization if any, optimizer algorithm used… Authors can choose to make a thorough description of the code or to publish it along the paper as a supplementary material.
Remark 16: The remark was not about precision; my question was about x2 coefficient being positive instead of negative. Machine learning methods are often black boxes meaning that we do not know what happens inside them during training. Therefore, it is essential to comment and discuss the results, especially if they do not follow the previously expected trends.
Additional remark: In line 503, Y seems to be already used as normalized data previously in the paper.
